# Closed magnetic topology in the Venusian magnetotail and ion escape at Venus

Shaosui Xu [1] ✉, David L. Mitchell[1], Phyllis Whittlesey [1], Ali Rahmati[1], Roberto Livi [1], Davin Larson[1], Janet G. Luhmann[1], Jasper S. Halekas[2], Takuya Hara[1], James P. McFadden[1], Marc Pulupa [1], Stuart D. Bale [1,3], Shannon M. Curry[1,4] & Moa Persson [5]

Venus, lacking an intrinsic global dipole magnetic field, serves as a textbook example of an induced magnetosphere, formed by interplanetary magnetic fields (IMF) enveloping the planet. Yet, various aspects of its magnetospheric dynamics and planetary ion outflows are complex and not well understood. Here we analyze plasma and magnetic field data acquired during the fourth Venus flyby of the Parker Solar Probe (PSP) mission and show evidence for closed topology in the nightside and downstream portion of the Venus magnetosphere (i.e., the magnetotail). The formation of the closed topology involves magnetic reconnection—a process rarely observed at non-magnetized planets. In addition, our study provides an evidence linking the cold Venusian ion flow in the magnetotail directly to magnetic connectivity to the ionosphere, akin to observations at Mars. These findings not only help the understanding of the complex ion flow patterns at Venus but also suggest that magnetic topology is one piece of key information for resolving ion escape mechanisms and thus the atmospheric evolution across various planetary environments and exoplanets.

Venus lacks substantial intrinsic global magnetic fields but possesses a thick and hot atmosphere[1,2]. Its ionosphere, a result of the upper atmosphere being photoionized by solar extreme ultraviolet (EUV) photons, is the main obstacle to the supersonic solar wind flow. This interaction induces currents in the conductive ionosphere that result in an induced magnetosphere, including a bow shock, a magnetosheath (MS) populated by shocked solar wind flow, a magnetic barrier from the interplanetary magnetic field (IMF) piling up and draping around the planet, and an unmagnetized or magnetized ionosphere depending on the solar conditions[3–5]. While Venus represents the most classical picture of an induced magnetosphere, various aspects of its magnetospheric dynamics and planetary ion escape are complex and not well understood.

Magnetic reconnection is an important process for efficient and fast conversion of magnetic energy into kinetic energy of plasma particles and also changing magnetic connectivity, operating in many astrophysical plasma environments. In the context of planetary plasma environments, magnetic reconnection is mainly expected to operate at magnetized planets, as a form of reconnection between the external magnetic fields and intrinsic magnetic fields, or self-reconnection of the intrinsic magnetic fields, such as Earth[6–8] and Mercury[9–11] or Mars with localized strong crustal magnetisms[12–16]. Despite Venus's magnetosphere being dominantly induced, evidence suggests magnetic reconnection is occurring at Venus as well. Reported relevant observations in the Venusian magnetotail include magnetic flux ropes[17,18], burst fluxes of escaping planetary ions[19], planetward flows[20], and in situ observations of the ion diffusion region[21]. Because of limited data

[1]Space Sciences Laboratory, University of California Berkeley, 7 Gauss Way, Berkeley, CA, USA. [2]Department of Physics and Astronomy, University of Iowa, Iowa City, IA, USA. [3]Physics Department, University of California Berkeley, 366 Physics North MC 7300, Berkeley, CA, USA. [4]Department of Astrophysical and Planetary Sciences, University of Colorado Boulder, 2000 Colorado Ave, Boulder, CO, USA. [5]Swedish Institute of Space Physics, Uppsala, Sweden. ✉e-mail: shaosui.xu@ssl.berkeley.edu

quality and coverage and the likely rarity of occurrence, we still know very little about the dynamics of magnetic reconnection and its role in reconfiguring the Venus magnetosphere.

While Venus's magnetosphere consists of piled-up IMF lines, the magnetic connectivity between the solar wind and the Venus ionosphere is, however, not as simple. Xu et al.[22] define and infer magnetic topology with respect to the Venus collisional atmosphere/ionosphere (~200 km altitude), rather than the planet's surface, with the super-thermal electron and magnetic field measurements from Venus Express (VEx)[23]: (a) draped topology with both ends of a field line connected to the solar wind without intersecting the ionosphere, (b) open topology with one end connected to the solar wind and the other to the ionosphere, and (c) closed topology with both ends embedded in the ionosphere. It is worth noting that these different types of magnetic topologies are mostly likely a result of the IMF penetrating to different depths of the ionosphere, differentiating if the field line has access to the main ionosphere. Xu et al.[22] reported an unexpected magnetic topology at Venus, cross-terminator closed field lines, in addition to the expected draped magnetic fields and open field lines (likely a result of IMF penetrating deep into the ionosphere). More recently, Xu et al.[24] statistically characterize the occurrence rates of different magnetic topologies at Venus and find that such cross-terminator closed topologies are not rare events but have an approximately 10% occurrence probability. The formation mechanism of such closed loops at the terminator for a planet without significant intrinsic fields is unknown and speculated to be associated with complex plasma flow in the ionosphere (giving rise to ionospheric currents), magnetic reconnection, or perhaps a weak intrinsic field, either from crustal sources or an internal dynamo. Xu et al. also statistically conclude the closed topology to mostly occur near the terminator and at low altitudes.

One important implication of magnetic connectivity to the ionosphere is that these field lines provide magnetic conduits between the ionosphere and the solar wind, along which cold ions can diffuse and escape, similar to the polar wind at magnetized planets[25,26]. At Mars, magnetic topology plays an important role in ion escape. Ions on draped magnetic field lines tend to be fast and low in density, mostly accelerated by the motional electric field ($\mathbf{E_M} = -\mathbf{U} \times \mathbf{B}$) and/or the $\mathbf{J} \times \mathbf{B}$ force[5,27], where $\mathbf{U}$ is the bulk flow velocity, $\mathbf{B}$ is the magnetic field vector, and $\mathbf{J}$ is the current density. In contrast, ions on open magnetic field lines tend to be slow, cold, and dense, and are mainly accelerated by the ambipolar electric fields arising from the electron pressure gradient[28–30]. At Venus, the connection between magnetic topology and ion escape is not yet clear. Planetary ions usually escape down the magnetotail, but sometimes they are observed to flow towards the planet[20,31–34], suggesting a more complicated picture for ion escape. The ion flow in the Venus magnetotail is highly responsive to the motional electric field, more energetic in the $+\mathbf{E_M}$ hemisphere, and the $\mathbf{J} \times \mathbf{B}$ force[5]. In the $-\mathbf{E_M}$ hemisphere, the ion flow pattern is much more irregular and is often observed to be planetward. Meanwhile, magnetic fields are more tightly wrapped in the $-\mathbf{E_M}$ hemisphere than the $+\mathbf{E_M}$ hemisphere[35], a configuration leading to more open (and perhaps closed) field lines. Magnetic topology is perhaps the missing piece of information for disentangling Venus's complex ion flow.

In this work, we present plasma and magnetic field observations made by the Parker Solar Probe during its fourth Venus flyby for a gravity assist, which provides a unique opportunity to investigate Venus's induced magnetosphere. By using electron pitch angle-energy measurements, we report a type of closed magnetic field at Venus whose formation is likely a result of magnetic reconnection in the magnetotail. In addition, our results suggest that most of this tail flyby consists of open and closed field lines populated by slow-moving, cold ions. Our study establishes a link between the cold Venusian ion flow in the magnetotail and its direct magnetic connectivity to the ionosphere. This finding has important implications for Venus's ion escape

and the atmospheric evolution of unmagnetized planets and exoplanets.

## Results
### Fourth Venus flyby observations of Parker Solar Probe
On 20 February 2021, PSP had its 4th Venus gravity assist flyby, during which PSP's trajectory was mainly in the $X_{VSO}$–$Y_{VSO}$ plane with near zero motion in the $Z_{VSO}$ direction and flew from [+$X_{VSO}$, −$Y_{VSO}$] to [−$X_{VSO}$, +$Y_{VSO}$], as shown in Fig. 1h, where VSO is the Venus-Solar-Orbital coordinates with its definition provided in the "Methods" section. Figure 1a–g shows the time series of the PSP observations. More specifically, PSP entered from the upstream solar wind to the magnetosheath on the −$Y_{VSO}$ side, crossing the bow shock at around 19:58 UT (universal time), signified by enhanced electron (Fig. 1c) and ion (Fig. 1e) fluxes and increased magnetic field strength (Fig. 1f). Closer in, at around 20:04 UT, the magnetic fluctuation amplitude (Fig. 1f)

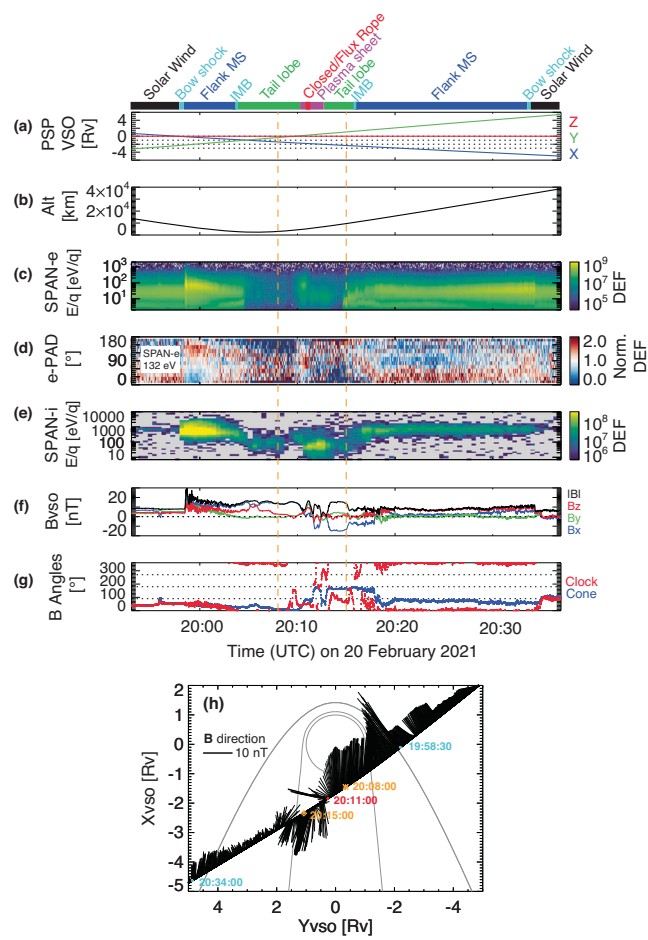

**Fig. 1 | Overview of the PSP observations on 20 February 2021, during its fourth Venus gravity assist.** Panels (**a**–**g**) are the time series of: **a** the PSP trajectory in the Venus-Solar-Orbital (VSO) coordinates, **b** the PSP altitude (km) relative to the Venus surface, **c** omnidirectional electron differential energy fluxes (DEF, eVcm⁻²s⁻¹sr⁻¹eV⁻¹) and **d** normalized 132-eV electron pitch angle distributions measured by the SPAN-e instrument, **e** ion differential energy fluxes (DEF, eVcm⁻²s⁻¹sr⁻¹eV⁻¹) averaged over all look directions measured by the SPAN-i instrument, **f** the 1-s magnetic field vector and strength and **g** magnetic clock angles $B_{clk}$ and cone angles $B_{cone}$ in the VSO coordinates, measured by the FIELDS instrument. The two vertical dashed lines highlight a zoomed-in time period shown in Fig. 2. Panel (**h**) shows the PSP trajectory in the $X_{VSO}$ − $Y_{VSO}$ plane, with the black whiskers displaying the $Bx_{VSO}$ and $By_{VSO}$ components. The gray lines show the empirical bow shock and the induced magnetosphere boundary (IMB)[63], in between is the magnetosheath (MS). The zoom-in period is indicated between the asterisk and diamond symbols.

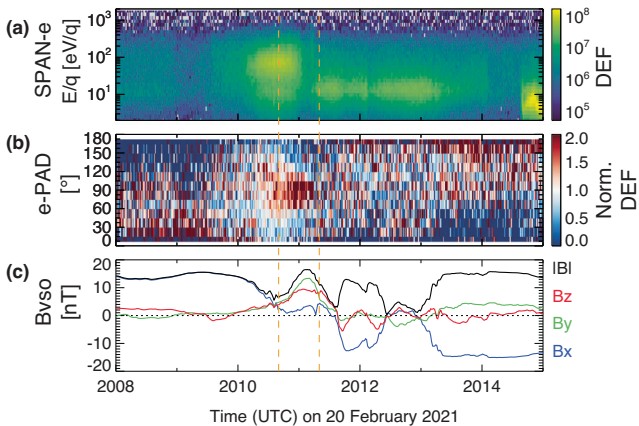

**Fig. 2 | The PSP electron and magnetic field observations zoomed in between 20:08 UT and 20:15 UT on 20 February 2021.** The three panels are time series of omnidirectional electron differential energy fluxes (DEF, eVcm$^{-2}$s$^{-1}$sr$^{-1}$eV$^{-1}$) (**a**) and normalized 132-eV electron pitch angle distributions (**b**) measured by the SPAN-e instrument, and **c** the 1-s magnetic field vector and strength in the VSO coordinates measured by the FIELDS instrument. The two vertical dashed lines highlight the time interval of two-sided loss cones.

becomes much reduced and the magnetic field strength increases, marking the crossing of the induced magnetosphere boundary, after which is the magnetic barrier (also the magnetotail lobe), where the IMF piles up and drapes around the planet.

As PSP entered the magnetotail lobe, ions measured by the SPAN-i instrument (Fig. 1e) center at much lower energies (<100 eV, not yet corrected for spacecraft potential or the spacecraft velocity of ~24 km/s) than those in the magnetosheath and the upstream solar wind. From 20:04 UT to 20:10 UT, the magnetic field (Fig. 1f) is around 15 nT and is mainly in the $-Bx_{VSO}$ direction, while from 20:13 UT to 20:15 UT, the magnetic field is mainly in the $-Bx_{VSO}$ direction with a magnitude of 15 nT. This is a typical crossing of two oppositely-pointed magnetotail lobes, which are separated by a plasma sheet. This particular plasma sheet between 20:10 UT and 20:13 UT is more structured than usual (for example, Fig. 2c of Collinson et al.[36]), embedded with ionospheric tail rays (narrow channels of cold plasma density enhancement coincident with low magnetic field magnitudes) centered around 20:11:30 UT and 20:12:30 UT, reported by Collinson et al.[37]. From 20:13 UT to 20:34 UT, PSP was traveling through the magnetosheath on the $-Y_{VSO}$ side and crossed the bow shock, and exited back into the solar wind at around 20:35 UT.

Throughout the orbit, the local magnetic clock angle $B_{clk}$ in panel (Fig. 1g) mainly varies between 0° (or 360°) and 90°, which reflects the upstream IMF conditions for an induced magnetosphere[38–40]. In particular, the magnetic field vector in the VSO coordinates immediately upstream of the bow shock is [4.63, 5.05, 3.94] nT with $B_{clk}$ = 52° for inbound (19:56:30–19:58:30 UT) and [−1.53, 6.19, 0.38] nT with $B_{clk}$ = 93° for outbound (20:35:00–20:36:00 UT). That is, the perpendicular component (w.r.t. $X_{VSO}$) of the upstream IMF is mainly in [$+By_{VSO}$, $+By_{VSO}$] before the encounter and mainly in $+Bz_{VSO}$ after. We use the average of the inbound and outbound upstream IMFs, [1.55, 5.62, 1.78] nT with $B_{clk}$ = 74°, to transform from the VSO coordinates to the VSE (Venus-Solar-Electric) coordinates for later discussion (see the definition of the VSE coordinate system in the "Methods" section)[41,42]. This is important for our interpretation of the data discussed in later sections.

The last highlight of Fig. 1 is the normalized electron pitch angle (PA) distribution in panel d, which shows electron flux depletion in the antiparallel direction (PA 134°–180°) for 20:05:10–20:09:30 UT and in the parallel direction (PA 0°–46°) for 20:13:10–20:15:50 UT. Combined with the information of the magnetic field direction (Fig. 1f), the

electron flux depletion in both periods occurs in the anti-Sunward (also anti-Venus-ward) direction, a typical signature for open field lines[22,43–49]. This type of pitch angle distribution is called a one-sided loss cone and the flux depletion is caused by sunward electrons precipitating along open field lines and being absorbed by the atmosphere via mainly neutral-electron collisions. That is, the magnetotail lobes for this particular flyby mostly consist of open magnetic field lines, in contrast to typical draped IMFs in the magnetotail[22].

**Electron pitch angle distributions and closed magnetic topology**
The time of interest of this study is highlighted between the two dashed vertical lines in Fig. 2, which corresponds to the time range marked by the two dashed vertical lines in Fig. 1 and only shows the electron data and magnetic field data. Particularly, as shown in Fig. 2b, unlike the one-sided loss cone pitch angle distributions within the two oppositely-pointing lobes or the more-or-less isotropic pitch angle distribution (PAD) bracketing the highlighted time range, electrons have the highest fluxes at perpendicular directions (PA 67°–113°) and the lowest fluxes at both parallel and antiparallel directions, or a so-called two-sided loss cone. This type of pitch angle distribution is normally caused by perpendicular electrons being trapped on closed magnetic fields and field-aligned electrons being absorbed by the atmosphere (with 10–30% backscattered)[43,44,50].

Magnetic reflection is independent of energy, so closed magnetic fields should cause two-sided loss cones over a wide range of energies. To verify, we select three cases (20:10:48–20:10:52, 20:10:57–20:11:01, and 20:11:11–20:11:15 UT) within the marked time interval in Fig. 2 and examine their electron pitch angle and energy distributions, as shown in Fig. 3. Panels a, d, and g of Fig. 3 are the normalized PADs by the averaged flux of each energy channel. This shows that high fluxes at the perpendicular PAs (bright pixels) occur across multiple energy channels from tens to hundreds of eV, spanning the widest range of 20–800 eV in panel g. PADs are more isotropic at lower energies, probably because of increased backscattering[50], possible field-aligned electrostatic potentials[22], and/or artificial effects (distorted electron trajectories because of the electromagnetic environment around the spacecraft or secondary electrons produced within the instrument). At high energies, the counting statistics are insufficient to provide meaningful PADs. Panels b, e, and h of Fig. 3 are the electron flux normalized to the maximum flux for four energy channels, from which the depletion of electron fluxes at parallel and antiparallel PAs is estimated to be 20–30%, consistent with values from Mars research[50]. Lastly, panels c, f, and i of Fig. 3 are the energy spectra separately for parallel (PA 0°–46°), perpendicular (PA 67°–113°), and antiparallel (PA 134°–180°) directions, which shows the perpendicular fluxes are highest compared to field-aligned directions.

The loss cone size within this period (indicated by the two vertical dashed lines in Fig. 2) is around 45°–60° from Fig. 3 and the magnetic strength is around 10–16 nT, which gives a magnetic strength of ~20 nT at the footpoints of closed magnetic field lines. This is in good agreement with the measured magnetic field strength by the Pioneer Venus Orbiter (PVO) on the nightside and at low altitudes (<200 km) near the footpoints of open/closed field lines[51], as shown in Supplementary Fig. 1. In addition, as shown in Fig. 2, the one-sided loss cones with a loss-cone size of roughly 60° occurring before (e.g., 20:08–20:09) and after (e.g., 20:13–20:15) the two-sided loss cones also have a local magnetic strength of about 15 nT, suggesting a similar magnetic strength at the footpoints as the closed loops. This is further evidence that the closed loop is likely a result of magnetic reconnection between two open-to-night field lines. Note that perpendicular heating of electrons, mostly by waves, could also cause higher fluxes at perpendicular PAs of particular energies[52]. In this case, no significant wave activities are observed and the trapped PADs are across a wide range of energies, both of which suggest that perpendicular heating is less likely to be the cause.

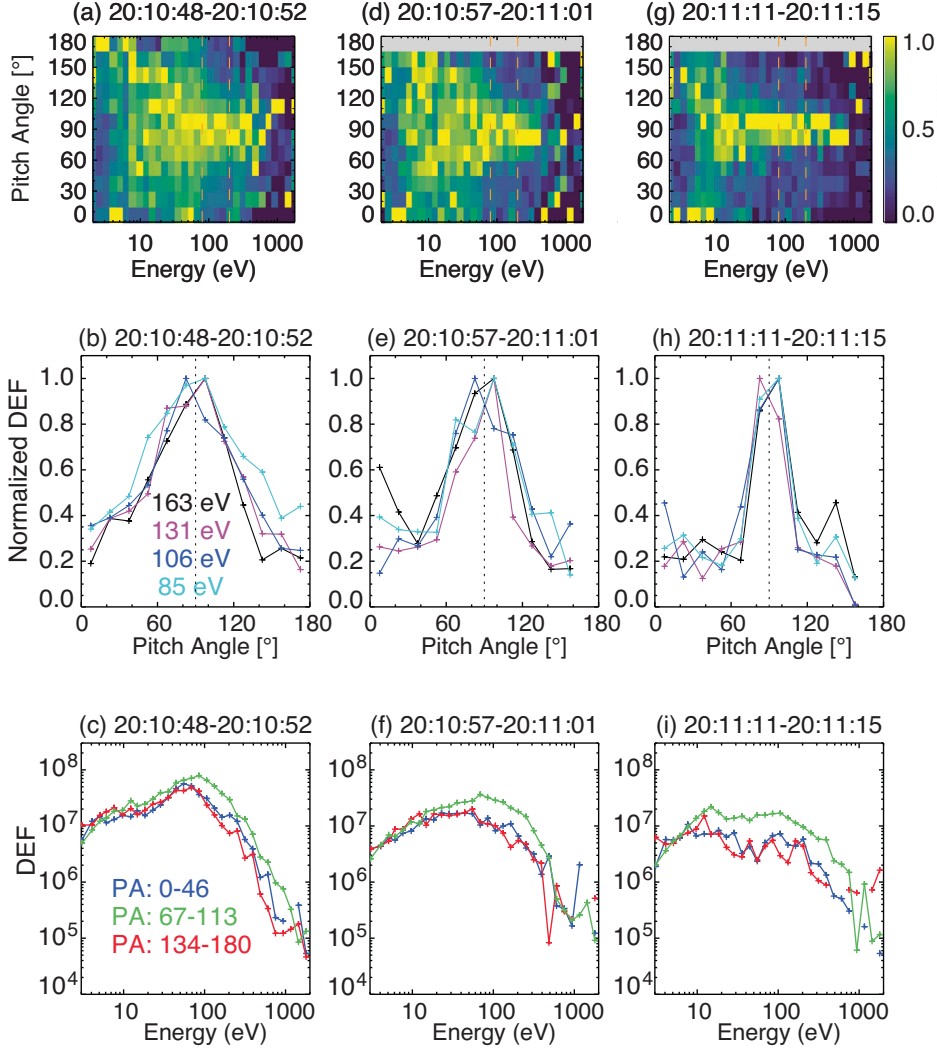

**Fig. 3 | Selected examples of electron pitch angle and energy distributions.**
Three selected times are 20:10:48–20:10:52 (panels **a**–**c**), 20:10:57–20:11:01 (panels **e**, **f**), and 20:11:11–20:11:15 UT (panels **g**–**i**), each averaging over 4-s SPAN-e observations. DEF stands for differential energy flux with a unit of eVcm$^{-2}$s$^{-1}$sr$^{-1}$eV$^{-1}$. Three rows are the normalized pitch angle distributions by the averaged flux of each energy channel (panels **a**, **d**, and **g**), normalized pitch angle distributions by the maximum flux of each energy channel within 80–200 eV (panels **b**, **e**, and **h**), and the energy distributions of electrons at parallel (PA 0°–46°), perpendicular (PA 67°–113°), and antiparallel (PA 134°–180°) directions (panels **c**, **f**, and **i**).

## Magnetic morphology and structure of the closed topology

To summarize the electron and magnetic field observations thus far, during this Venus flyby, PSP encountered different plasma regimes as it crossed the Venusian tail, and various magnetic topologies were inferred from the observations. As noted above, the magnetic clock angle $B_{clk}$ in the upstream and magnetosheath for both inbound and outbound suggests the perpendicular component of the upstream IMF would likely be mainly [$+By_{VSO}$, $+Bz_{VSO}$]. It is informative to show the magnetic vectors along the PSP trajectory in the VSE coordinates in Fig. 4a–c, using the averaged upstream IMF direction mentioned above. In VSE, the X direction is opposite to the solar wind flow and the Y axis is along the IMF component perpendicular to X such that the Z direction is along the motional electric field of the solar wind flow. PSP was crossing the induced magnetosphere from [$+X_{VSE}$, $-Y_{VSE}$, $+Z_{VSE}$] to [$-X_{VSE}$, $+Y_{VSE}$, $-Z_{VSE}$] (Fig. 4a–c). The yellow dots in all panels mark the position of the closed topology, before and after which are two oppositely-directed lobes (Fig. 4a). Note that the tail current sheet (yellow dots) is located slightly eastward at $Y_{VSE} \approx +0.5R_V$ also shown in previous observations and simulations[53,54].

In Fig. 4d–e, we illustrate the magnetic topology encountered by PSP in the $X_{VSE} - Y_{VSE}$ and $Y_{VSE} - Z_{VSE}$ planes, respectively. In between

two oppositely-directed lobes (Fig. 4b) consisting of mainly open field lines, embedded in the highly-structured plasma current sheet is a short period (~40 s) of closed magnetic topology. An interesting point here is that, if we draw a simple closed loop, as indicated by the dotted red line in Fig. 4d, the magnetic vector would be mainly in the $-Y_{VSE}$ direction, having to be opposite to the upstream IMF (intrinsically $+By_{VSE}$) based on typical draping geometries. If the upstream IMF was mainly [$+By_{VSO}$, $+Bz_{VSO}$] as suggested by $B_{clk}$ in the upstream and magnetosheath, the simple closed loop should have the opposite magnetic field direction to the upstream at the apex of the loop, [$-By_{VSO}$, $-Bz_{VSO}$] in this case. This, however, contradicts the observations of mainly [$+By_{VSO}$, $+Bz_{VSO}$] within this period (Fig. 2c).

To reconcile this discrepancy, we propose two scenarios. Scenario 1 would still be the simple closed loop scenario (the dotted red line with the yellow dot marked with 1 in Fig. 4d) but the perpendicular component of the upstream IMF during this period would have to be [$-By_{VSO}$, $-Bz_{VSO}$], in contrast to an IMF with [$+By_{VSO}$, $+Bz_{VSO}$] before and after. It is worth noting that under such a scenario, the upstream IMF would switch between opposite orientations at least a couple of times, which enables magnetic reconnections between oppositely draped IMFs, similar to the scenario proposed by Edberg et al.[55] under severe

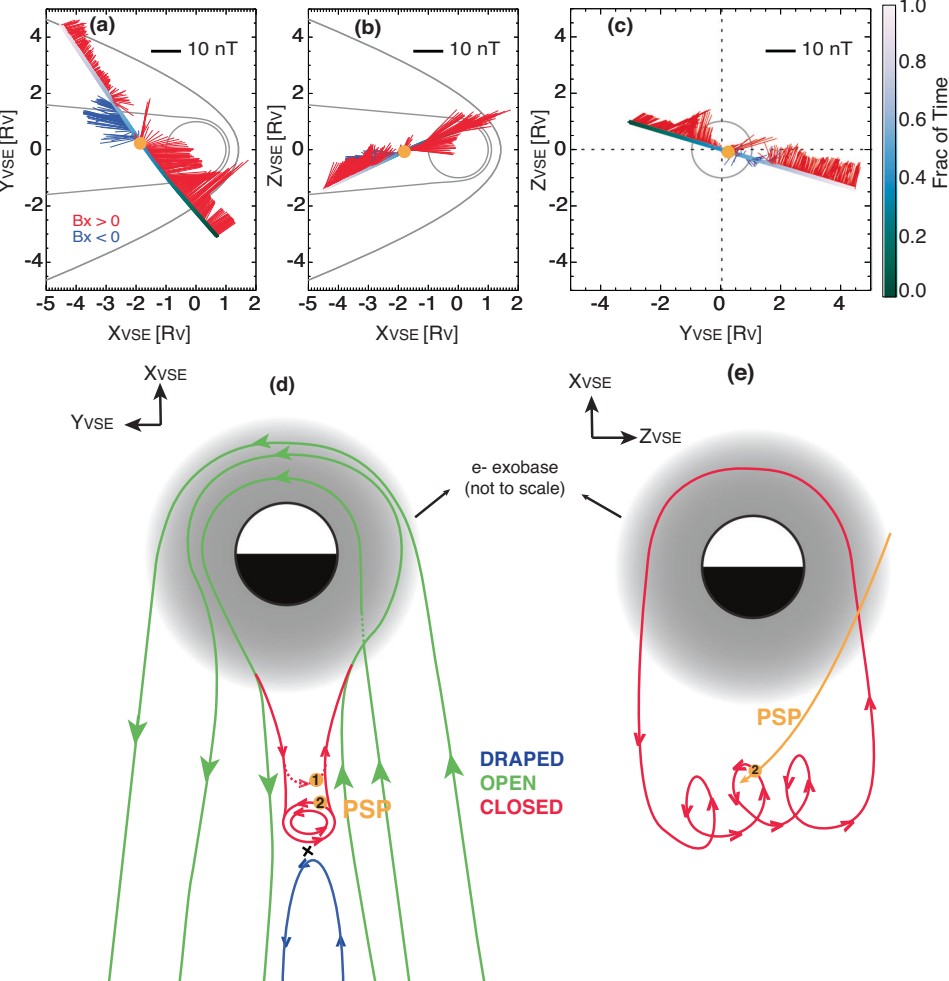

**Fig. 4 | Schematics of the PSP locations and how PSP transverses the closed field structure.** Panels (**a**–**c**) show the magnetic field components measured by PSP in the VSE coordinates in three projections, **a** $Bx_{VSE}$ − $By_{VSE}$ in the $X_{VSE}$ − $Y_{VSE}$ plane, **b** $Bx_{VSE}$ − $Bz_{VSE}$ in the $X_{VSE}$ − $Z_{VSE}$ plane, **c** $By_{VSE}$ − $Bz_{VSE}$ in the $Y_{VSE}$ − $Y_{VSE}$ plane, blue and red for negative and positive $Bx_{VSE}$. The cartoon in panel (**d**) illustrates the tail magnetic topology for this PSP Venus flyby in the $X_{VSE}$ − $Y_{VSE}$ plane. In particular, the

red lines show two scenarios of the closed topology: (1) a simple closed loop (dashed) and (2) a flux rope connected to the ionosphere (solid). The cartoon in panel (**e**) illustrates the ionosphere-closing flux rope in the $Y_{VSE}$ − $Z_{VSE}$ plane, with the PSP trajectory shown as the yellow line. The yellow dots in all panels mark the location of PSP encountering the closed topology.

space weather conditions. In fact, the upstream IMF before and after the encounter was quite structured, as shown in Supplementary Fig. 2, such that it is possible to have large rotations in IMF within tens of minutes.

Meanwhile, if the upstream IMF only varied between $+By_{VSO}$ and $+Bz_{VSO}$ when PSP was inside of the Venus magnetosphere, then the apex of the closed topology is the same as the upstream IMF. For this scenario (Scenario 2), we invoke a magnetic structure, a magnetic flux rope closing to the collisional atmosphere, as indicated by the solid red lines in Fig. 4d with the yellow dot marked with 2. Magnetic flux rope is a helical structure with a strong core field and is generally character-ized by an enhancement in the magnetic field strength and magnetic field reversal if the spacecraft transverses through the center of the structure. Magnetic flux ropes can be a result of magnetic reconnec-tion or plasma instability and have been identified in both the Venus ionosphere[56,57] and magnetotail[17,18]. In this case, PSP observed an increase in the magnetic field strength but not the field reversal (Fig. 2c), which means PSP might only cross the edge of the structure (yellow dot 2 in Fig. 4d). As PSP did not cut through the center, the minimum variance analysis[58] gives inconclusive results regarding the exact configuration of the flux rope (Supplementary Fig. 3). None-theless, the helical structure of a magnetic flux rope can also explain a

closed topology having the same apex as the upstream IMF as shown in Fig. 4d. In Fig. 4e, we sketch the magnetic flux rope in the $Y_{VSE}$ − $Z_{VSE}$ plane with an axis mainly along the $+Z_{VSE}$ axis (an idealized config-uration). The magnetic flux rope with a closed topology would likely also be a result of magnetic reconnection in the magnetotail rather than plasma instability (no obvious association with closed topology). Lastly, the main factor to differentiate the two scenarios is the real-time upstream IMF condition, which was not available with only single-spacecraft observations but requires multi-spacecraft observations at Venus.

## Cold planetary ion flow and magnetic topology

In this section, we investigate the possible relationship between the ion behavior and magnetic topology during this tail flyby. Figure 5 shows electron energy spectra and PADs in panels a, b and the ion energy and mass-per-charge (m/q) spectra in panels c and d. The mass-per-charge spectra (panel d) show that the dominant ion species is $H^+$ of a solar wind origin (i.e., the shocked solar wind ions) before 20:05 UT and after 20:17 UT while planetary heavy ions, mainly $O^+$ and $O_2^+$, dominate between 20:05 UT and 20:16 UT. Note that light ions, likely $H^+$ and $H_2^+$, within 20:05 UT–20:16 UT, could also be of planetary origin.

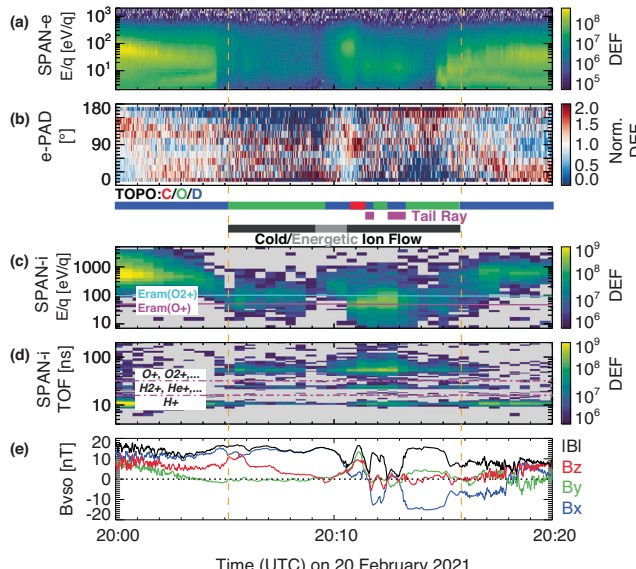

**Fig. 5 | The PSP electron, ion, and magnetic field observations zoomed in between 20:00 UT and 20:20 UT on 20 February 2021.** Panels (**a**–**e**) are time series of: omnidirectional electron differential energy fluxes (DEF, eVcm$^{-2}$s$^{-1}$sr$^{-1}$eV$^{-1}$) (**a**) and normalized 132-eV electron pitch angle distributions (**b**) measured by the SPAN-e instrument, ion differential energy fluxes (DEF, eVcm$^{-2}$s$^{-1}$sr$^{-1}$eV$^{-1}$) (**c**) and ion mass spectra (in DEF, eVcm$^{-2}$s$^{-1}$sr$^{-1}$eV$^{-1}$) averaged over all looking directions (**d**) measured by the SPAN-i instrument, and **e** the 1-s magnetic field vector and strength in the VSO coordinates measured by the FIELDS instrument. Between panels (**b**) and (**c**) are color bars for magnetic topology (red/green/blue for closed/open/draped, respectively), tail rays, and cold ion flow.

The ion energy spectra in Fig. 5c are overlaid with the ram ion energies for $O^+$ ($E_{ram}(O^+)$, magenta line) and $O_2^+$ ($E_{ram}(O_2^+)$, cyan line), i.e., the energy gain from spacecraft velocity. The ram energy is relatively high because PSP was traveling at a speed of ~24 km/s along the $[-X_{VSO}, +Y_{VSO}, 0]$ direction. Between 20:05 UT and 20:16 UT, the energy of peak ion fluxes is very close to $E_{ram}(O^+)$ and $E_{ram}(O_2^+)$ and the ions are mainly from the ram direction, suggesting these ions have an insignificant bulk velocity compared to the spacecraft speed. Unfortunately, we cannot accurately calculate the ion moments because of spacecraft potentials unavailable and insufficient information contained in the downlinked data products. We can, however, infer the ions during 20:05–20:09 UT and 20:10–20:16 UT (as indicated by black bars above Fig. 5c) to be slow (order of 10 km/s or less) and cold (order of 10 eV). Meanwhile, there is an energy-dispersion-like structure (the energy of peak ion fluxes varying from a few hundred eV to about 100 eV) between 20:09:10–20:10:30 UT (gray bar above Fig. 5c), resembling typical ion energization across a tail current sheet[31].

Magnetic topology inferred from electron pitch angle and energy distributions is shown as the color bar right below Fig. 5b, red/green/blue for closed/open/draped, respectively. By comparing magnetic topology and ion observations, there is a good agreement between time periods of cold planetary ion flow and open/closed magnetic topologies while the draped field lines concur with energetic ions, except for two tail rays[37] as indicated by the purple bars between Fig. 5b and 5c. We also show another example of tail observations from VEx in Supplementary Fig. 4, where both protons and heavy ions are hot and/or energized when the tail lobes consist of mainly draped field lines. These case studies suggest a strong connection between magnetic topology and tail ion dynamics.

That is, our study provides an evidence linking the cold ion outflow/inflow at Venus directly to magnetic connectivity (both open and closed topologies) with its ionosphere. At Mars, because of the

magnetic connectivity to the ionosphere, ions on open and closed field lines tend to be both cold and denser, as ions of a higher density require more energy to get energized and the $J \times B$ force is also not as prominent on these topologies as on draped field lines. Such an explanation is also applicable to this Venus case study. Notably, if the closed topology indeed represents a magnetic flux rope closing to the ionosphere, the associated ions exhibit characteristics of being slow and cold, in contrast to the energized ions suggested by Zhang et al. and Dubinin et al.[17,19]. This difference could be because the magnetic flux rope reported in our study is connected to the collisional atmosphere, as opposed to the detached magnetic flux rope reported by Zhang et al.[17].

## Discussion

This study presents plasma and magnetic field observations from PSP during its 4th Venus gravity assist flyby on 20 February 2021. As PSP transversed the Venusian magnetotail, its electron observations suggest that the magnetotail lobes mainly consist of open field lines connected to the nightside collisional atmosphere. Embedded in the highly structured plasma sheet that separates the two lobes, there is a brief time period (~40 s) of closed magnetic topology, an unexpected configuration at Venus. While Xu et al.[22] reported an evidence of closed magnetic topology at Venus, it was only observed near the terminator, and its formation mechanisms still remain unknown. In contrast, here we report a closed magnetic topology in the magnetotail, which is likely a result of magnetic reconnection in the magnetotail. We propose two possible configurations for the closed topology, simple closed loops associated with rapid upstream IMF rotations or a part of a magnetic flux rope that closes to the collisional atmosphere. To differentiate these two possible configurations requires multi-spacecraft observations at Venus. There has been very limited research on magnetic reconnection at Venus, due to limited observations and/or a very low-occurring phenomenon at an unmagnetized planet. Here, PSP provided an uncommon observation of lobes mostly consisting of open field lines, instead of the typical dominant draped field lines[24], and the plasma sheet is highly structured and embedded with tail rays[37], in addition to possible highly variable upstream IMF orientations. All these factors could contribute to the triggering of the magnetotail magnetic reconnection between the opposite-pointing open magnetic field lines in the two lobes.

Apart from the closed topology, the direct magnetic connectivity to the ionosphere (both open and closed) has important implications for planetary ion escape. The ion observations from PSP indicate that, during this flyby, the magnetotail is mainly populated with planetary ions, which are slow and cold. The pattern of cold ion flow is consistent with direct magnetic connectivity to the ionosphere. It suggests that, just like at Mars, the ion behavior is well organized by magnetic topology in the tail, particularly in terms of cold ion flow. Our study suggests that Venus's complex ion flow pattern is perhaps in part a result of various magnetic topologies in the tail. In particular, the closed topology might be one possible scenario to explain the ion return flow, of which the physical driving mechanism is yet to be discovered. Meanwhile, magnetic flux ropes tend to be associated with bursty fluxes of escaping planetary ions, leading to enhanced ion escape[19]. Another highlight from this study is that our results reveal a possible magnetic flux rope configuration that is closing to the ionosphere, which is populated with slow and cold ion flow, not necessarily leading to largely enhanced ion outflow as the detached magnetic flux rope[17].

To summarize, our study reveals closed magnetic loops in the magnetotail, of which the formation mechanism is likely magnetic reconnection, a process not frequently observed at unmagnetized planets. In addition, our research suggests that magnetic topology plays an important role in organizing ion flow within Venus's tail, just as at Mars. Our findings help the understanding of the complex ion flow

patterns at Venus and thus the Venus ion escape. It also suggests that magnetic topology is one piece of key information for resolving ion escape mechanisms and thus the atmospheric evolution across various planetary environments and exoplanets.

## Methods

### Observations

In this study, we use superthermal electrons observations from the Solar Probe ANalyzer for Electrons (SPAN-e)[59], ion observations from the Solar Probe ANalyzer - Ions (SPAN-i)[60], and magnetic field observations from the FIELDS magnetometer instrument[61] onboard PSP. The SPAN-e instrument provides electron measurements at an energy range of 2 eV–30 keV with 32 energy steps with a $\Delta E/E = 16.7\%$, with a field of view of $240° × 120°$ at an angular resolution of 6° or 24° and a measurement cadence of 0.435 s. The SPAN-i instrument provides compositional ion measurements at an energy range of 2 eV–30 keV (6 eV–20 keV for this encounter) with 32 energy steps with a $\Delta E/E = 16.7\%$, with a field of view of $247.5° × 120°$ at an angular resolution of 11.25° or 22.5° and a measurement cadence of 0.435 s. The main SPAN-i data products include three ion mass bins: $H^+$, $H_2^+$ or $He^+$, heavy ions. The magnetic field vectors are measured by the outboard magnetometer as a part of the FIELDS instrument suite at a time cadence of 292.97 Sa/s with a measurement dynamic range of ±65536 nT.

### Venus-solar-orbital (VSO) coordinates

$X_{VSO}$ points from the center of Venus to the Sun, $Z_{VSO}$ points to the ecliptic north pole of Venus' orbit plane, and $Y_{VSO}$ completes the right-handed system.

### Magnetic clock and cone angles in VSO

We define a magnetic clock angle $B_{clk}$ as $B_{clk} = \tan^{-1}(By_{VSO}/Bz_{VSO})$ and the magnetic cone angle $B_{cn}$ as $B_{cn} = \cos^{-1}(Bx_{VSO}/|B|)$.

### Venus-solar-electric (VSE) coordinates

$X_{VSE}$ is opposite to the solar wind flow, the $Y_{VSE}$ is in the direction of the perpendicular component of the IMF (with respect to $X_{VSE}$), and $Z_{VSE}$ is in the same direction as $\mathbf{E_M} = -\mathbf{U} \times \mathbf{B}$.

### Normalized pitch angle distribution

For all electron pitch angle distributions presented in this study, the electron fluxes at different pitch angles are normalized for each energy separately for every single measurement. The normalization is either by the averaged flux or the maximum flux across all pitch angles at each energy channel.

## Data availability

All data used in this paper are public. The PSP data is publicly archived at https://spdf.gsfc.nasa.gov/pub/data/psp/. FIELDS data can be found at https://spdf.gsfc.nasa.gov/pub/data/psp/fields/. SPAN-e data can be found at https://spdf.gsfc.nasa.gov/pub/data/psp/sweap/spe/. SPAN-i can be found at https://spdf.gsfc.nasa.gov/pub/data/psp/sweap/spi/. The source data files for the figures in this current study have been deposited in a Zenodo repository (https://doi.org/10.5281/zenodo.11404158). The datasets generated during and/or analysed during the current study are available from the corresponding author upon request. Source data are provided with this paper.

## Code availability

Data access and processing was done using SPEDAS V3.1 (http://spedas.org/wiki/index.php?title=Downloads_and_Installation)[62].

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

## Acknowledgements

We acknowledge support from the Parker Solar Probe (PSP) mission and the FIELDS and Solar Wind Electrons, Alphas, and Protons (SWEAP) teams through contract NNN06AA01C. PSP was designed, built, and is now operated by the Johns Hopkins Applied Physics Laboratory as part of NASA's Living with a Star (LWS) program. S.X. acknowledges support from NASA's Solar System Working Program, grant #80NSSC21K0151.

## Author contributions

S.X. conceived the study and wrote the initial draft of the paper. S.X., D.M., P.W., A.R., R.L., D.L., J.L., J.H., T.H., J.M., and M. Persson carried out the data analysis and interpretation. S.X., D.M., P.W., A.R., R.L., D.L., J.H., T.H., J.M., M. Pulupa, S.B., S.M., and M. Persson have read and provided feedback on the paper. P.W. is the instrument lead of the PSP SPAN-e instrument, R.L. is the instrument lead of the PSP SPAN-i instrument, and D.L. and A.R. are Co-Is of SPAN-i and SPAN-e. S.B. is the instrument lead of the PSP FIELDS instrument and M. Pulupa is a Co-I of FIELDS.

## Competing interests

The authors declare no competing interests.
