## [Peer Review File · Nature Communications]

1. We have created a transparent peer review file for your submission, please confirm if the figures on pages 13, 15, 16, 26, 28 can be published. We can remove them if they infringe copyright.

I confirm that they can be published.

2. Please provide an 'Equally Contributing' Statement in your affiliations, that is connected to a symbol in your author list.

- It is only necessary to provide an email address for your corresponding author.

The corresponding author is highlighted as "*" in the manuscript and the email address is shaosui.xu@ssl.berkeley.edu. Below is the complied manuscript. I'm not sure how to make it clearer.

*Corresponding author(s). E-mail(s): shaosui.xu@ssl.berkeley.edu;

Contributing authors: mitchell@ssl.berkeley.edu;
phyllisw@berkeley.edu; rahmati@ssl.berkeley.edu;
rivi@berkeley.edu; davin@berkeley.edu;
jgluhman@ssl.berkeley.edu; jasper-halekas@uiowa.edu;
hara@ssl.berkeley.edu; mcfadden@ssl.berkeley.edu;
pulupa@berkeley.edu; bale@berkeley.edu;
shannon.curry@colorado.edu; moa.persson@irf.se;

3. Please ensure all your main figures have a figure title.

We have provided the figure title as the bolded sentence in the figure caption for each figure.

4. Please cite all authors in your author contributions at least once. Only 12 of 14 authors have been cited.

This has been corrected.

5. Please use the following headings for the Supplementary Information: Supplementary Methods, Supplementary Discussion, Supplementary Notes, Supplementary References. The current headings which do not fall under those categories can be subheadings.

- Please also provide your supplementary references at the end of your supplementary information file.

We have provided a new Supplementary Information document that satisfies these requirements.

6. Please label and cite supplementary figures and tables as 'Supplementary Figure X' and 'Supplementary Table X', respectively (in both the main text and the Supplementary Information file).

We have revised the manuscript and the SI file accordingly.

7. Panel 'f' appears to be cited twice in the legend for 'Supplementary Figure 4'. Please amend the second citation to 'g'.

Corrected. Thanks for catching that.

8. Abstract: Please replace "the first" with "an".

Revised as suggested.

9. Abstract: Please note that your abstract would benefit from clarifying the third sentence " Through plasma...magnetosphere). We recommend "Here we analyze plasma and magnetic field data acquired during the fourth Venus flyby of the Parker Solar Probe (PSP) mission and show evidence for closed topology at the Venus magnetotail in the nightside and downstream portion of the magnetosphere."

We have changed it to "Here we analyze plasma and magnetic field data acquired during the fourth Venus flyby of the Parker Solar Probe (PSP) mission and show evidence for closed topology in the nightside and downstream portion of the Venus magnetosphere (i.e., the magnetotail)."

10. Introduction, paragraph 3: Please revise to avoid overlap with <https://agupubs.onlinelibrary.wiley.com/doi/10.1029/2023JA032133>

We have simplified it to "Xu et al. [2023][24] statistically characterized the occurrence rates of different magnetic topologies at Venus using VEx measurements and found closed topology to mostly occur near the terminator and at low altitudes." Hopefully this separates the two studies.

11. Main text: Please remove "first", "new", "first time".

Revised as suggested.

12. Figure 1: Please leave space between panels. Y-axis titles are not accessible. Please include a title (Observation date and time) below panel g.

Edited as suggested.

13. Figure 2: Please include a title below panel c.

Edited as suggested.

14. Figure 3: Please leave vertical space between panels. Dates above the panels and x-titles are not accessible.

Edited as suggested.

15. Figure 5: Please include a title below figure 5

Edited as suggested.

16. Data availability: Please include a statement about the availability of the data generated in this study. Please refer to <https://www.nature.com/documents/nr-data-availability-statements-data-citations.pdf>

The source files used to produce the figures are available in a Zenodo repository (<https://doi.org/10.5281/zenodo.11404158>), as stated in the Data Availability section. I'm not sure what else to provide. I have added a sentence at the end stating: "Source data are provided with this paper."

17. Please remove "Inclusion & ethics statement". For the article sections, please see <https://www.nature.com/documents/ncomms-formatting-instructions.pdf>

Removed as suggested.

18. Please remove, "first time", "first evidence", "new".

Edited as suggested.

19. Methods: Please ensure to provide sufficient information such that the experiments can be reproduced without reference to other papers.

We have provided sufficient information regarding the data used for this study in this section. We included references to the instruments/experiments in this section to give credits/references to the instrument papers and also in case readers want to learn more about the instruments.

20. Methods, instruments: We believe "observations" title is more representative for this section. If there are specific software related to the data, please include here.

Changed as suggested.

21. Editorial question: Could you please clarify whether magnetic topology is solely based on observations? If some modelling, estimates, extrapolations are involved, please explain them in the methods.

Yes, magnetic topology is solely based on superthermal electron and magnetic field observations and no modeling is involved. This is a commonly used method at various planetary objects (Mars, Venus, Moon, Titan, etc) and various observational data (MAVEN, Venus Express, THEMIS-ARTEMIS, Cassini, etc).

22. Supplementary Information contains data from Pioneer Venus Orbiter (PVO) data and Venus Express's (VEx). Please add data links to the Data Availability section and include statements about their availability. Please ensure to provide sufficient information such that the experiments can be reproduced without reference to other papers.

We have added statements about the data availability used in the Supplementary Information in the main text and also the SI document itself.

23. Supplementary information: Please clarify how the hodograms were generated.

Hodograms are just the plots of B_i vs B_j and B_i vs B_k , based on the minimum variance analysis (MVA). We have edited the figure to not use this phrase but only MVA throughout the file to avoid confusion.